# Role of Spinal Cholecystokinin Octapeptide, Nociceptin/Orphanin FQ, and Hemokinin-1 in Diabetic Allodynia

**DOI:** 10.3390/biomedicines12061332

**Published:** 2024-06-15

**Authors:** Takafumi Hayashi, Syu-ichi Kanno, Chizuko Watanabe, Damiana Scuteri, Yasuyuki Agatsuma, Akiyoshi Hara, Giacinto Bagetta, Tsukasa Sakurada, Shinobu Sakurada

**Affiliations:** 1Division of Pharmaceutics, Faculty of Pharmaceutical Sciences, Tohoku Medical and Pharmaceutical University, Sendai 981-8558, Japan; thayashi@tohoku-mpu.ac.jp (T.H.); agatsuma@tohoku-mpu.ac.jp (Y.A.); 2Division of Clinical Pharmacotherapeutics, Faculty of Pharmaceutical Sciences, Tohoku Medical and Pharmaceutical University, Sendai 981-8558, Japan; syu-kan@tohoku-mpu.ac.jp (S.-i.K.); hara-a@tohoku-mpu.ac.jp (A.H.); 3Division of Physiology and Anatomy, Faculty of Pharmaceutical Sciences, Tohoku Medical and Pharmaceutical University, Sendai 981-8558, Japan; w-chizu@tohoku-mpu.ac.jp; 4Department of Health Sciences, University “Magna Graecia” of Catanzaro, 88100 Catanzaro, Italy; damiana.scuteri@unicz.it; 5Pharmacotechnology Documentation & Transfer Unit, Department of Pharmacy, Preclinical & Translational Pharmacology, Health & Nutritional Sciences, University of Calabria, 87036 Rende, Italy; g.bagetta@unical.it; 6Faculty of Pharmacy, Daiichi University of Pharmacy, Fukuoka 815-8511, Japan; tsukasa@daiichi-cps.ac.jp; 7Faculty of Pharmaceutical Sciences, Tohoku Medical and Pharmaceutical University, Sendai 981-8558, Japan

**Keywords:** cholecystokinin octapeptide, nociceptin/orphanin FQ, hemokinin-1, diabetic allodynia, spinal cord

## Abstract

A complication of diabetes is neuropathic pain, which is difficult to control with medication. We have confirmed that neuropathic pain due to mechanical allodynia in diabetic mice is mediated by a characteristic neuropeptide in the spinal cord. We evaluated the strength of mechanical allodynia in mice using von Frey filaments. When mice were intravenously injected with streptozotocin, mechanical allodynia appeared 3 days later. Antibodies of representative neuropeptides were intrathecally (i.t.) administered to allodynia-induced mice 7 days after the intravenous administration of streptozotocin, and allodynia was reduced by anti-cholecystokinin octapeptide antibodies, anti-nociceptin/orphanin FQ antibodies, and anti-hemokinin-1 antibodies. In contrast, i.t.-administered anti-substance P antibodies, anti-somatostatin antibodies, and anti-angiotensin II antibodies did not affect streptozotocin-induced diabetic allodynia mice. Mechanical allodynia was attenuated by the i.t. administration of CCK-B receptor antagonists and ORL-1 receptor antagonists. The mRNA level of CCK-B receptors in streptozotocin-induced diabetic allodynia mice increased in the spinal cord, but not in the dorsal root ganglion. These results indicate that diabetic allodynia is caused by cholecystokinin octapeptide, nociceptin/orphanin FQ, and hemokinin-1 released from primary afferent neurons in the spinal cord that transmit pain to the brain via the spinal dorsal horn.

## 1. Introduction

Neuropathic pain is excruciating and can be caused by diabetes, genetic diseases, tumors, nerve injury, cancer chemotherapy, and shingles [1]. Diabetic mechanical allodynia is a neuropathic pain that is one of the major complications of diabetes mellitus, resulting from a diffuse symmetric injury to the peripheral nerves that is associated with hyperglycemia [2,3]. Allodynia reduces quality of life and the self-care behaviors necessary for diabetes care, which is a disincentive for exercise therapy. Attempts have been made using many drug therapies to treat diabetic mechanical allodynia, but they have failed to control pain symptoms [4]; so, new and effective drug therapies are needed.

Although diabetic mechanical allodynia occurs as an abnormality in sensory processing, it is possible that the threshold of the Aδ/C fibers in primary sensory nerves is reduced or that Aβ fibers transmit pain via the pain pathway. In addition, the increased activity of glial cells in the spinal cord plays an important role in the development and maintenance of diabetic allodynia by causing the overexcitation of spinal dorsal horn neurons [5].

Substance P and calcitonin gene-related peptide (CGRP) are representative neuroactive peptides that reach the nerve endings through axonal transport from the dorsal root ganglia; they are stored in synaptic vesicles and are released into the spinal dorsal horn. Other neuropeptides involved in pain transmission, such as somatostatin, cholecystokinin octapeptide, nociceptin/orphanin FQ, angiotensin II, spermine, and hemokinin-1, are present in sensory nerves [6]. Spermine is synthesized in glial cells as well as neurons in the spinal cord and affects excitability and calcium influx [7,8]. Hemokinin-1 was first found in mouse bone marrow cells but has since been reported to be distributed in the whole body as well as in the central nervous system [9,10].

Antagonists of the neuropeptide receptors are known to attenuate allodynia and hyperalgesia caused by neuropathy. There are two receptor subtypes on which cholecystokinin octapeptide acts: the CCK-A (CCK1) receptors, which are distributed primarily in the periphery, and the CCK-B (CCK2) receptors, which are mainly distributed in the central nervous system [11,12]. Diabetes-induced hyperalgesia is due to the activation of protein kinase C in the spinal cord via CCK-B receptors [13]. Allodynia caused by spinal cord injury is attenuated by antagonists of the opioid receptor-like 1 (ORL-1) receptor on which nociceptin/orphanin FQ acts [14]. Hemokinin-1 is partially increased in activated microglial cells in the spinal cord in the early phase after sciatic nerve injury and is related to the development of neuropathic pain [15]. Hemokinin-1 is also involved in pain transmission through its involvement in neuron–glia interactions in the context of pathology and inflammation in the trigeminal nervous system and spinal cord [16,17]. It is possible that diabetes causes an increase in certain spinal neuropeptides, which activates the receptors on these neuropeptides, resulting in the development of allodynia.

Allodynia caused by neuropathy is a sustained excitation from the injured sensory nerve that is transmitted to the central nervous system via neuropeptides in the spinal cord. Therefore, in this study, we focused on pain-related neuropeptides in the spinal cord during diabetic allodynia using a diabetic mouse model created by administering streptozotocin (Table 1), and we investigated their behavioral pharmacological effects using the von Frey method. We then determined which neuropeptides were involved in pain transmission in the spinal cord during diabetic allodynia.

## 2. Materials and Methods

This experiment was conducted with the consent of the Animal Experiment Committee of Tohoku Medical and Pharmaceutical University. Male ddY mice (Japan SLC, Hamamatsu, Japan) weighing 22–26 g were used for the experiments. These mice are a non-inbred strain maintained as a closed colony in Japan and are suitable for experiments with diabetic allodynia due to their streptozotocin sensitivity. The mice were kept in a 12 h light–dark cycle (light period 7:00–19:00, dark period 19:00–7:00) at a constant room temperature of 23 ± 1 °C and humidity of 52 ± 2% and were fed solid samples (FR-2: Oriental Yeast, Tokyo, Japan) and tap water ad libitum except during the experiment. The total number of mice used in this entire study was 270, and all the mice used were single-use.

The following chemicals were used: streptozotocin (Sigma-Aldrich, Burlington, MA, USA); anti-CCK 8, rabbit-poly (Immunostar, Hudson, WI, USA); anti-nociceptin, rabbit-poly (Novusbio, Littleton, CO, USA); anti-hemokinin-1 (ms, rt), rabbit-poly (BMA Biomedicals, Basel, Switzerland); anti-angiotensin II, mouse-mono (Bertin Pharma, formerly SPI Bio, York, UK); anti-somatostatin, rabbit-poly (Gene Tex, Irvine, CA, USA); anti-substance P, mouse-mono (R&D Systems, Minneapolis, MN, USA); anti-spermidine, rabbit-poly (Affinity Biosciences, Melbourne, Australia); anti-spermine, rabbit (Gene Tex); SR27897 (Tocris, Bristol, UK); CI988 (Tocris); and JTC-801 (Chem Scene, Monmouth, NJ, USA). The methods of administration and the dosage of each substance are shown in Table 2.

An amount of 200 mg/kg of streptozotocin dissolved in saline solution (Otsuka Pharmaceutical Factory, Tokushima, Japan) was injected intravenously into the tail vein of mice to induce significant mechanical allodynia [18]. The streptozotocin solution should be prepared fresh and injected within 5 min of being dissolved. After the mice were administered streptozotocin, blood glucose was measured to determine whether diabetes had accurately developed. Mice with blood glucose levels of 300 mg/dL or higher after 72 h of streptozotocin administration were considered to have developed diabetes. Allodynia developed from days 3 to 14 after streptozotocin administration, and the mice recovered after about 42 days (Figure 1). In this study, the von Frey filament technique was performed using a mouse model of diabetic neuropathic pain 7 days after the intravenous administration of 200 mg/kg of streptozotocin by means of tail injection (Figure 2).

Each reagent administered into the spinal subarachnoid space was dissolved and diluted in artificial cerebrospinal fluid (CSF: Tocris). Intraspinal subarachnoid administration was performed at a rate of 5 μL/mouse using a 29-gauge injection needle (Hoshiseido Medical Instrument Industry, Tokyo, Japan) attached to a 50 μL microsyringe (Hamilton, Reno, NV, USA). Agents were administered promptly and without anesthetic into the spinal subarachnoid space between lumbar vertebrae 5 and 6 after the mice were firmly immobilized with fingers to prevent movement, in accordance with Hylden and Wilcox’s method [21].

We used von Frey filaments (Neuroscience, Tokyo, Japan) of different stimulus intensities (0.02, 0.04, 0.07, 0.16, 0.4, and 0.6 g), to which a defined pressure could be applied, and we placed one mouse at a time into a cage with a wire mesh bottom to fully acclimate them to the measurement environment. The von Frey filaments were pressed vertically onto the foot pads of the left hindlimbs of the mice, and we observed whether this evoked an escape response in the mice from the 5 s stimulation by the filament. When an escape response was observed, the thinnest filament was used, and when no escape response was observed, the thickest filament was used. The intensity in grams of the thinnest filament that elicited an escape response was evaluated as the pain threshold. One group comprised 10 animals, and the mean and standard error of the pain thresholds were calculated.

The messenger RNA (mRNA) levels of the CCK-A and -B receptors in the lumbar spinal dorsal horn and dorsal root ganglion (DRG) were measured through real-time qPCR assays. A group of six animals was tested. After the mice were decapitated, the mouse L3–L5 lumbar spinal dorsal horn and DRG were promptly removed. The extraction of total RNA was carried out by using the RNA extraction reagent KOD SYBR qPCR Mix (Toyobo, Osaka, Japan). The mRNA levels of the CCK-A and -B receptors were calculated using the 2—(target Ct − reference Ct) formula. β-actin functioned as the reference gene. The primer sequences were (5′-3′): CCK-A–TGTTGCCCGAATCCTGGTC (forward primer) and GGTGCTCATGTGGCTGTAGGAA (reverse primer); CCK-B–CCCATAGCCTAGTGCGGTAGTGA (forward primer) and GAGATCTGCGGCTCCGAAAG (reverse primer) (Takara-bio, Kusatsu, Japan).

All data are presented as calculated means and standard errors. Statistical analyses were performed using GraphPad Prism 8.4.3 (686, GraphPad Software, Boston, MA, USA). For comparisons between the two groups, Dunnett’s or Sidak’s multiple comparison test was performed at the 5% level of significance after one-way or two-way analysis of variance. A difference was considered significant if the risk rate was less than or equal to 5%.

## 3. Results

### 3.1. Effects of Anti-Neuropeptide Antibodies Administered to the Spinal Cord with Streptozotocin-Induced Neuropathic Pain on Pain Threshold

Anti-neuropeptide antibodies were administered intrathecally (i.t.) to mice with diabetic neuropathy and their pain thresholds were then tested after anti-neuropeptide antibody administration via the von Frey test (Figure 3). The pain threshold represents the minimum stimulus intensity at which pain is perceived, with an increase in the threshold resulting in less pain being felt and vice versa. No change in the pain threshold was observed after the i.t. administration of anti-angiotensin II antibodies, anti-somatostatin antibodies, anti-substance P antibodies, anti-spermidine antibodies, and anti-spermine antibodies 1:12.5 (12.5× dilution) (Figure 3d–h). The pain thresholds were significantly elevated from 30 min after the i.t. administration of anti-cholecystokinin octapeptide antibodies, anti-nociceptin/orphanin FQ antibodies, and anti-hemokinin-1 antibodies (Figure 3a–c). Comparison of the pain thresholds 60 min after administration of concentrations of anti-cholecystokinin octapeptide antibodies, anti-nociceptin/orphanin FQ antibodies, and anti-hemokinin-1 antibodies showed no increase in the pain threshold at low concentrations of each antibody (1:100), but at 1:25 and 1:50 of each antibody, the pain thresholds returned to those before streptozotocin administration.

### 3.2. Effects of Antagonists for the CCK-A, CCK-B, and Nociceptin/Orphanin FQ Receptors (ORL-1) Administered to the Spinal Cord with Streptozotocin-Induced Neuropathic Pain on Pain Threshold

Changes in the pain threshold were measured using the von Frey method in mice with diabetic neuropathy after the i.t. administration of CCK-A receptor antagonist, SR27897, and CCK-B receptor antagonist, CI988 (Figure 4). The diabetic mice had no change in their pain thresholds 240 min after the i.t. administration of SR27897. In contrast, the i.t. administration of CI988 to diabetic mice resulted in a significant dose-dependent increase in the pain threshold, with a peak at 45 to 60 min. The significant increase in the pain threshold with the i.t. administration of CI988 lasted from 30 to 150 min after administration, and the pain thresholds returned to pre-CI988 values at 180 to 240 min.

JTC-801, a nociceptin/orphanin FQ receptor (ORL-1) antagonist, was i.t. administered to mice with diabetic neuropathy, and the time course of the pain threshold was measured immediately after administration (Figure 5). A sustained increase in the pain threshold was observed from 45 to 120 min after the administration of JTC-801.

### 3.3. Alterations in Cholecystokinin Receptor mRNA Levels in Spinal Dorsal Horn and DRG Neurons in Diabetic Allodynia

To determine whether the cholecystokinin receptors in mice are altered by diabetes, the level of cholecystokinin receptor mRNA in the dorsal root ganglia (DRG) and spinal dorsal horn of the streptozotocin-induced mice was measured via real-time qPCR assays. The mRNA level of CCK-B receptors in mice increased significantly in the lumber spinal dorsal horn, but not the DRG of streptozotocin-induced diabetic allodynia (Figure 6). The mRNA level of CCK-A receptors in the lumber spinal dorsal horn and DRG did not change with diabetic allodynia.

## 4. Discussion

In this study, we sought to identify the neuropeptides that are involved in pain transmission in the spinal cord of mice that develop diabetes-induced allodynia. Some studies have investigated the behavioral pharmacology of neuropeptide antibodies administered directly into the subarachnoid space of the spinal cord to suppress the function of neuropeptides in the spinal fluid [22,23]. To this end, antibodies against neuropeptides were administered directly into the spinal cord of mice to investigate their effects on diabetes-induced allodynia. The administration of anti-cholecystokinin octapeptide antibodies, anti-nociceptin/orphanin FQ antibodies, and anti-hemokinin-1 antibodies to the spinal cord of mice with diabetes induced by streptozotocin significantly restored mechanical allodynia. In each of these cases, allodynia was attenuated in a concentration-dependent manner with each antibody diluted, suggesting that the pathological pain caused by streptozotocin administration was attenuated by the action of each antibody. However, the administration of anti-substance P antibodies, anti-somatostatin antibodies, anti-angiotensin II antibodies, anti-spermine antibodies, and anti-spermidine antibodies to the spinal cord did not show any effect on mechanical allodynia. In contrast to these results, rats modeling neuropathy produced by sciatic nerve cuff implantation demonstrated increased amounts of substance P in the spinal cord [24]. It was suggested that there may be differences in the development of mechanical allodynia depending on the cause of neuropathy.

Cholecystokinin octapeptide is a neuropeptide originally discovered as a gastrointestinal hormone, which is involved in pain transmission in the spinal cord [25]. In the present study, the mice that were intrathecally administered anti-cholecystokinin octapeptide antibodies exhibited significantly attenuated allodynia in an antibody concentration-dependent manner. Thus far, two subtypes of cholecystokinin receptors (CCK-A and CCK-B) have been reported [26,27]. The spinal administration of cholecystokinin octapeptide into the rostral ventromedial medulla of naïve rats produces a robust tactile allodynic effect and a modest hyperalgesia [28]. The intrathecal administration of cholecystokinin octapeptide in naïve mice induces nociceptive behaviors such as licking the lower abdomen and biting the lower legs and tail [25]. Changes in the cholecystokinin octapeptide system have also been associated with neuropathic pain, as peripheral nerve injury results in an elevated level of mRNA for cholecystokinin octapeptide, as well as CCK-A and CCK-B receptors in the dorsal root ganglia [29,30].

In diabetes, pain sensitivity is enhanced by cholecystokinin octapeptide, and the mechanism is mediated by CCK-B, but not CCK-A, receptors located on the primary afferent neurons [31]. In the present study, the mice developed diabetic allodynia and were not affected in any way by the administration of a CCK-A receptor antagonist. In contrast, the administration of a CCK-B receptor antagonist significantly attenuated allodynia in a dose-dependent manner. In addition, it has been suggested that diabetes-induced hyperalgesia or allodynia in mice could be due in part to the enhanced release of cholecystokinin octapeptide in the spinal cord [13]. Thus, it is likely that increased cholecystokinin octapeptide content and/or activity in diabetes might cause pathological changes in nociceptive transmission in primary afferent neurons. Diabetes causes the inhibition of protein kinase C delta phosphorylation in neurons such as the dorsal root ganglia, slowing cell growth [32]. In the present study, we found that streptozotocin administration increased mRNA levels of CCK-B receptors in the spinal dorsal horn, but not in the dorsal root ganglia. It is suggested that cholecystokinin octapeptide is not increased by neuronal growth retardation, due to hyperglycemia, and therefore the expression of CCK-B receptors is not affected in the dorsal root ganglia.

Nociceptin/orphanin FQ is an endogenous ligand for opioid receptor-like 1 (ORL-1), which resembles the structure of opioid receptors and is a neuropeptide that transmits pain [33,34,35]. Diabetes-induced neuralgia is more pronounced in ORL-1 receptor knockout mice than in other opioid receptor knockout mice [36]. The ORL-1 receptor antagonist is also effective for neuropathic and inflammatory pain by acting on the spinal cord [14]. It has also been reported that chronic neuropathy and diabetes increase nociceptin levels in the brain and spinal cord, supporting the results of this study, in which anti-nociceptin antibodies administered to the spinal cord attenuated diabetic allodynia [37]. The present study did not examine changes in the mRNA levels of ORL-1 receptors in the spinal dorsal horn and the dorsal root ganglia during the process of diabetic allodynia. ORL-1 receptor mRNA has been identified in the spinal cord and dorsal root ganglia [38], but immunohistochemical staining studies have shown a weaker distribution of ORL-1 receptors in the dorsal root ganglia and in lamina II to III of the spinal dorsal horn [39]. In diabetic allodynia, it may be difficult to confirm changes in the mRNA levels of ORL-1 receptors in the spinal cord and dorsal root ganglia.

Hemokinin-1 is involved in the transmission of pain and itching, which are classified in the tachykinin family as substance P [10]. Hemokinin-1 was suggested to be involved in the transmission of diabetes-induced mechanical allodynia, albeit weakly compared to cholecystokinin octapeptide and nociceptin/orphanin FQ. Hemokinin-1 is involved in pain transmission during neuropathic and inflammatory pain [17]. Interestingly, the tachykinin family, substance P, showed no involvement in diabetes-induced mechanical allodynia.

We found that hemokinin-1 is involved in diabetes-induced allodynia development. Hemokinin-1 plays an important role in neuropathic and arthritis pain, which is triggered by several mechanisms [17,40]. In contrast, substance P is not involved in diabetes-induced allodynia, even though it belongs to the tachykinin family and acts on NK1 receptors such as hemokinin-1. Hemokinin-1, depending on the dose, produces the nociceptive behaviors via NMDA receptors, but not NK1 receptors in the spinal cord [41]. Hemokinin-1 may be involved in the development of diabetes-induced allodynia via a pathway not involving traditional NK1 receptors. It has been suggested in reports that there are either specific receptors for hemokinin-1 or that there are signaling pathways that are distinct from substance P [42,43].

We found that when anti-angiotensin II antibodies were intrathecally administered to mice with diabetic allodynia and the mice were then examined for changes in actual pain-related behaviors, no effect was observed. However, it has been reported that angiotensin II is involved in neuropathic pain in studies of diabetic allodynia such as ours. The immunofluorescence intensity for angiotensin II in the spinal dorsal horn of streptozotocin-treated mice after 14 days was markedly increased compared to vehicle mice [44]. Since we examined the neuropeptides involved in spinal pain transmission in allodynia expressed 7 days after streptozotocin administration to mice, we suspect that this longitudinal difference may be related to the involvement of angiotensin II.

## 5. Conclusions

Diabetic allodynia developed in mice from day 3 after the administration of streptozotocin; the strongest allodynia was on day 7, and it persisted significantly until day 28. The mechanical allodynia expressed by streptozotocin-induced neuropathy is mediated by cholecystokinin octapeptide and nociceptin/orphanin FQ in the spinal cord. Hemokinin-1 is weakly but significantly involved in the development of diabetic allodynia, but the involvement of neuropeptides involved in pain transmission, such as substance P of the same tachykinin family, was precluded in the spinal cord. Since this study was performed in mice, it could not be argued that neuropeptides are involved in the spinal cord during the development of diabetic allodynia in humans in the same way that they are in mice. It is hoped that treatments targeting the actions of these neuropeptides will provide a stepping stone to overcome neuropathy in diabetes.

## Figures and Tables

**Figure 1 biomedicines-12-01332-f001:**
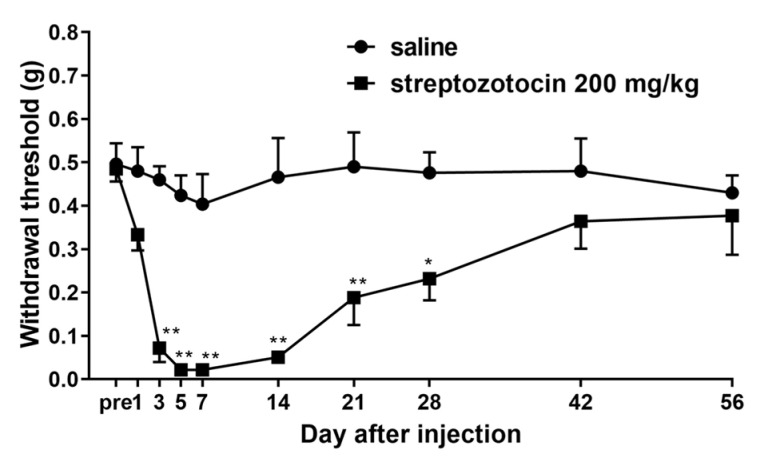
Changes over time in mechanical allodynia induced by intravenous streptozotocin in mice. Pain thresholds (withdrawal threshold; grams) were measured each time using von Frey filaments. Closed squares indicate 200 mg/kg of streptozotocin administered intravenously; closed circles indicate 10 mL/kg of saline administered intravenously. Each value represents the mean ± S.E.M. of 10 mice in each group. ** *p* < 0.01 and * *p* < 0.05 when compared with saline using Sidak’s multiple comparisons test.

**Figure 2 biomedicines-12-01332-f002:**
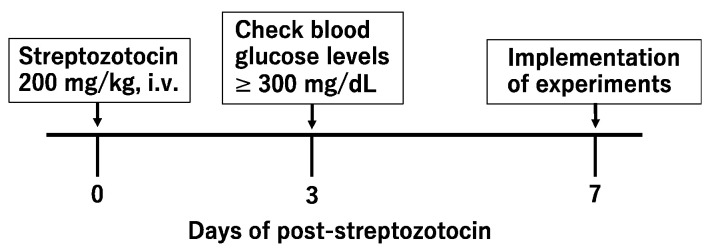
Diagram of the experiment.

**Figure 3 biomedicines-12-01332-f003:**
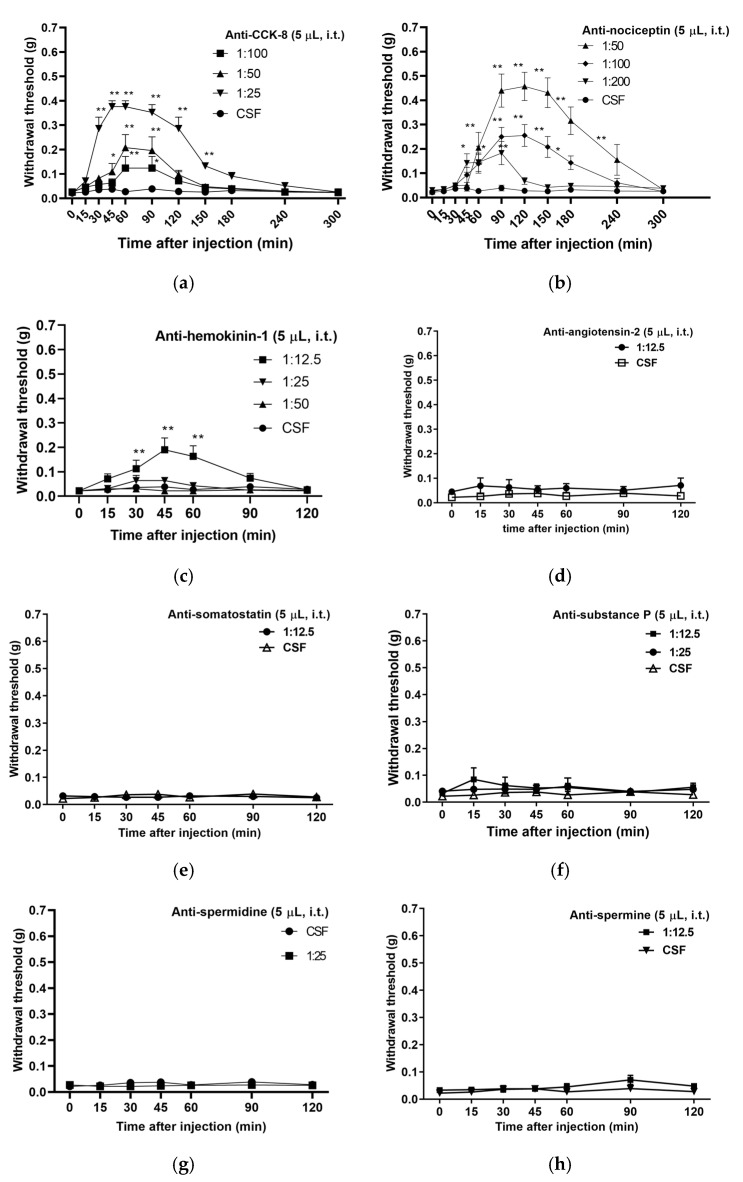
Effects of anti-neuropeptide antibodies administered to the spinal cord of mice with streptozotocin-induced diabetic neuropathy. Pain thresholds were measured via the von Frey method immediately after i.t. administration of (**a**) anti-cholecystokinin octapeptide (CCK-8) antibodies, (**b**) anti-nociceptin/orphanin FQ (nociceptin) antibodies, (**c**) anti-hemokinin-1 antibodies, (**d**) anti-angiotensin II (angiotensin-2) antibodies, (**e**) anti-somatostatin antibodies, (**f**) anti-substance P antibodies, (**g**) anti-spermidine antibodies, and (**h**) anti-spermine antibodies. Each value represents the mean ± S.E.M. of 10 mice in each group. ** *p* < 0.01 and * *p* < 0.05 when compared with the CSF control using Dunnett’s multiple comparisons test.

**Figure 4 biomedicines-12-01332-f004:**
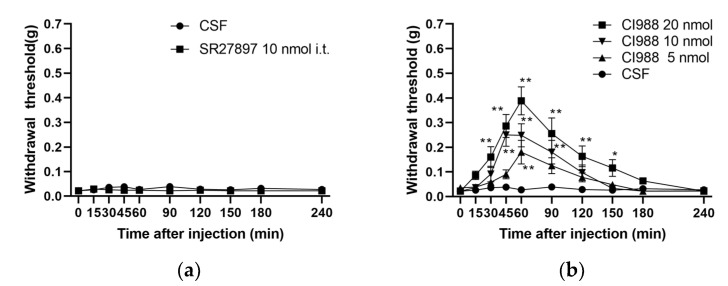
Effects of antagonists on the CCK-A and CCK-B receptors administered to the spinal cord of mice with streptozotocin-induced diabetic neuropathy. Pain thresholds were measured using the von Frey method immediately after i.t. administration of each antagonist ((**a**) SR27897, CCK-A receptor antagonist, (**b**) CI988, CCK-B receptor antagonist). Each value represents the mean ± S.E.M. of 10 mice in each group. ** *p* < 0.01 and * *p* < 0.05 when compared with the CSF control using Dunnett’s multiple comparisons test.

**Figure 5 biomedicines-12-01332-f005:**
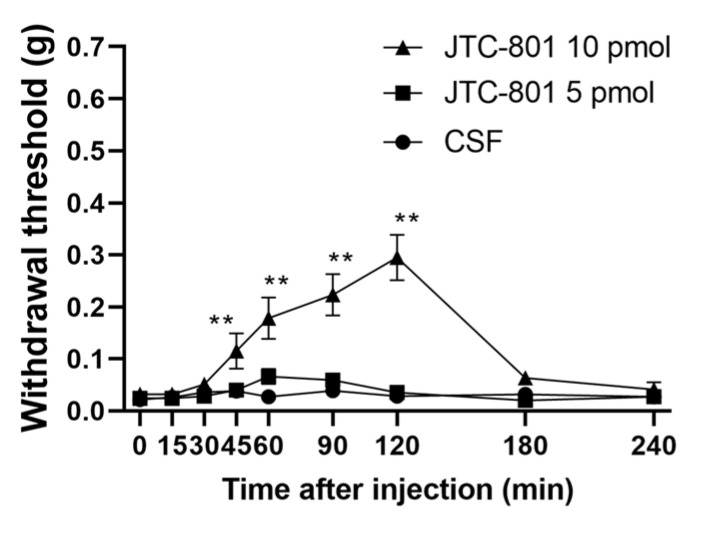
Effects of antagonists for the nociceptin/orphanin FQ receptor (ORL-1 receptor) administered to the spinal cord of mice with streptozotocin-induced diabetic neuropathy. Pain thresholds were measured by the von Frey method immediately after the i.t. administration of JTC-801 (a nociceptin/orphanin FQ receptor; ORL-1 receptor antagonist). Each value represents the mean ± S.E.M. of 10 mice in each group. ** *p* < 0.01 when compared with the CSF control using Dunnett’s multiple comparisons test.

**Figure 6 biomedicines-12-01332-f006:**
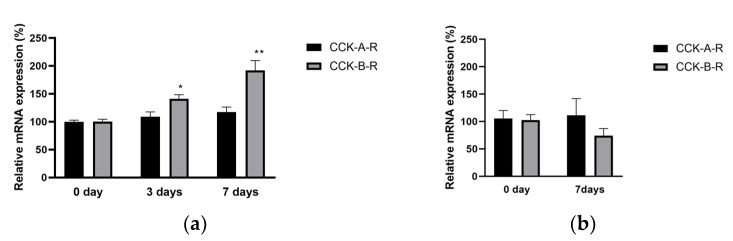
Changes over time in the percentages of CCK-A and CCK-B receptor mRNAs in (**a**) the lumber spinal dorsal horn and (**b**) the lumber dorsal root ganglia of mice with streptozotocin-induced diabetic neuropathy compared to day zero. Black columns represent the percent change in CCK-A receptor (CCK-A-R) mRNA levels. Gray columns represent the percent change in CCK-B receptor (CCK-B-R) mRNA levels. Each value represents the mean ± S.E.M. of 6 mice in each group. ** *p* < 0.01 and * *p* < 0.05 when compared with the CSF control using Dunnett’s multiple comparisons test.

**Table 1 biomedicines-12-01332-t001:** Various information about streptozotocin-induced type 1 diabetic mice.

Objectives of the Experiment	Mouse Streptozotocin Treatment	Reference Number
Model of diabetes mellitus	i.v.^1^ or i.p.^2^ 200 mg/kg (single, high doses)	[18]
Antidiabetic effect of S-8300 (a peptide extracted from shark liver)	i.v., 150 mg/kg (single, high doses)	[19]
Diabetic atherosclerosis models	i.p. 40 mg/kg (5 consecutive days, low doses)	[20]

^1^ Intravenous injection. ^2^ Intraperitoneal injection.

**Table 2 biomedicines-12-01332-t002:** Method and dosage of administration of the substances.

Administered Substances	Method of Administration	Doses
streptozotocin	i.v. ^1^	200 mg/kg
anti-neuropeptide antibody	i.t. ^2^	1:12.5–1:200
SR27897	i.t.	10 nmol
CI988	i.t.	5–20 nmol
JTC-801	i.t.	5–10 nmol

^1^ Intravenous injection. ^2^ Intrathecal injection.

## Data Availability

The data presented in this study are available within the article.

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
