# Peer review of "Role of Spinal Cholecystokinin Octapeptide, Nociceptin/Orphanin FQ, and Hemokinin-1 in Diabetic Allodynia"

_biomedicines, 2024, doi:10.3390/biomedicines12061332_

Round 1

Reviewer 1 Report (Previous Reviewer 1)

Comments and Suggestions for Authors

The authors reviewed and corrected the manuscript and added the data requested by the reviewer. I believe that now the article meets the conditions to be published.

Author Response

Reviewer 2 Report (Previous Reviewer 2)

Comments and Suggestions for Authors

While the manuscript improved in some aspects, some critical concerns were not addressed. For example, the capability of antibody as antagonist is not proved. How could authors be sure that these antibodies can work as blockers, especially for those show no effect? Authors should list literatures if there are experiments that show it works, or prove it if none.

Author Response

Reviewer 3 Report (New Reviewer)

Comments and Suggestions for Authors

Thank you for permitting me to review this manuscript

In this experimental animal rats study , the authors created diabete allodynia by injecting  IV streptozotocin this decreased the pain threshold  measured by von frei filament on tails

this was followed by  intrathecal injection of  different  anti neuropetides antibodies which attenuated allodynia In contrast, intrathecal.-administered anti-substance P antibodies, anti-somatostatin antibodies, and anti-angiotensin II anti-bodies did not affect streptozotocin-induced diabetic allodynia mice

Please declare the main objectives of the study at the end of the introduction

Its not clear how manu mice was used in this experimental study , page 4 the authors state that there was 10 animal , is that the real number ? if yes please state it at the start of method section

Is there any piblication on the effect of IV administration of these antipeptides instead of IT ? ,in other words how necessary a control group with IV injection could strengthen the eresults

Since this is a animal study the authors  should precise and moderate their conclusions since there is no validation in humans yet

Round 2

Reviewer 2 Report (Previous Reviewer 2)

Comments and Suggestions for Authors

Accept in present form

Author Response

Thank you for taking time out of your busy schedule to review our manuscript. You have helped us to improve our manuscript. We learned a lot from you.

This manuscript is a resubmission of an earlier submission. The following is a list of the peer review reports and author responses from that submission.

Round 1

Reviewer 1 Report

Comments and Suggestions for Authors

Observations on the article biomedicines-2565457

Abstract

“i.t.” must first be put in parentheses after the word “intrathecally”.

Not protein antibodies, but anti-protein antibodies.

Introduction

It is said about spermine that it is produced by both neurons and glial cells. We understand from this that the rest of the analyzed peptides are produced only by neurons. But hemokinin-1 was first described after it was discovered in B lymphocytes.

The two receptors, CCK-B and ORL-1 receptors, should also be mentioned here, what their role is and why they were chosen for the investigations in this study.

A diagram of the experiment, a summary figure is required.

Materials & Methods

 Explain why the ddY strain mice was chosen.

How many animals were used in the 2 groups, treatment with STZ and controls.

Why the dose of 200 mg/kg was chosen and in what concentration, for the onset of diabetes. What is the literature consulted or the experiments that led to the choice of this unique dose.

It would have been good to test the level of insulin in the blood.

Von Frey filament tests were too briefly described. How many seconds was the filament applied, at what time interval, how was the nociceptive response calculated?

The method of administration of the substances, peptides and anti-neurotransmitter antibodies, must be entered here, not in the Results. It is necessary a table with all the administered substances, the method of administration, in what doses, at what intervals.

A very good example of how to test mice with induced diabetes and how to write a good Materials and Methods chapter can be viewed from DOI: 10.3389/fphar.2020.628438.

Statistical analysis is missing.

Discussions

The authors showed data about CCK from the literature, but no comparison is made with their results. The fragment looks more like a presentation for the Introduction, than a discussion of the results obtained compared to the data from the literature.

Regarding hemokinin-1 vs SP, the data from the literature, DOI: 10.1016/j.npep.2016.12.003 (to be added to the bibliography), specify a different or even opposite action and it is suggested that there is a specific receptor for hk-1 or a different signaling pathway compared with SP. So the results obtained by the authors confirm this fact.

The authors showed that no involvement of spinal angiotensin II in diabetes-induced allodynia, after 7 days from STZ treatment, although the literature data reported an increased angiotensin II level after 14 days from STZ treatment in mice. If there are studies that show this fact, why weren't tests done after 14 days in this experiment as well? How can we still believe that the results are conclusive, if more tests were not done at longer time intervals? I suggest redoing the experiment and testing more at longer time intervals.

Conclusions

The conclusions are too concise. Although the discussions are not very well done either, the original results obtained by the authors should be highlighted more in the conclusions.

The study is a valuable one, the idea is good, but the results must be improved.

I suggest consulting more data from the literature, e.g only 3 articles about Hk-1:

doi: 10.1038/bjp.2008.301

doi: 10.3390/ijms21082938

doi:10.1016/j.brainresbull.2019.01.015

Comments on the Quality of English Language

There are some minor mistakes of expression in English that need to be corrected.

Reviewer 2 Report

Comments and Suggestions for Authors

This manuscript by Hayashi and colleagues presents a study on the role of spinal peptides in the maintenance of diabetic allodynia. The diabetic animal model was induced by streptozotocin. Antibodies and/or pharmacologic blockers against individual peptides/receptors are used to detect their role in the maintenance of diabetic allodynia. qPCR was performed to measure the mRNA level of cholecystokinin receptors in both spinal cord and DRGs of diabetic mice. While this work explored these peptides/receptors’ role in diabetic neuropathic pain, several serious flaws limit the possibility of publishing the paper in Biomedicine.

1. Several studies has explored the roles of nociceptin and cholecystokinin B receptor in diabetic neuropathic pain, which diminishes its novelty.

2. The manuscript needs to be significantly improved.

I) The grammar and syntax need to be improved. There are grammar/syntax errors throughout the manuscripts. For example (but not limited to):

 a. In abstract: “Spinal histamine was also caused by the development of allodynia”

 b. In method section: “Each reagent administered into the spinal subarachnoid space dissolved and diluted in artificial cerebrospinal fluid (CSF: Tocris) was administered at a rate of 5 μL/mouse by inserting a 29-gauge needle (Hoshiseido Medical Instrument Industry, Tokyo, Japan) at-tached to a 50 μL micro syringe (Hamilton, Nevada, USA) into the intrathecal space be-tween lumbar vertebrae Nos. 5 and 6 under anesthesia”

c. In result section: “Comparison of pain thresholds at 60 min post-administration by concentration showed no increase in pain threshold at low concentrations (1:100), but at 1:25 and 1:50, pain thresholds returned to the pain threshold value before streptozotocin administration”

II) The rigor of the study needs to be improved

a.   A naïve animal should be included to see whether the treatment affect the baseline, which can tell whether the effect observed due to its effect on physiological or pathological functions.

b.   Use of antibodies as blockers is complicate. The antibody must be able to affect its agonist-receptor binding site or change the conformation of the proteins. How could authors be sure that these antibodies can work as blockers, especially for those show no effect? Authors should list literatures if there are experiments that show it works, or prove it if none.

c.  While the level of CCK receptors were detected by qPCR, the level of nociceptin/orphanin FQ receptor mRNA expression in spinal dorsal horn and DRG was not mentioned although its antibodies/blockers decreased diabetic allodynia. Authors should explain why only CCK was detected in this study. 

III) Discussion needs to be improved

a.   Authors recognized the literatures that CCK and CCK-B are increased in primary sensory neurons from diabetic rats, which is different from this study performed with ddY mice. What causes these differences? This need to be discussed.

b.   In clinic, females reported greater pain intensity. However, the sex of animal is not indicated in the study. If one sex of animal is used in the study, could the result be the same in another sex of animal?

Minor issues:

1.   The primer sequences used in Real time PCR should be designated as forward and reverse primers. It is difficult to make out which is forward and which one is reverse.

2.   The graphical representation of Figure 2 should be improved. Minutes written in X axis of Panel (b) and (d) is not clear. Panel a, b, c, d, are large while the rest of them are small. Authors should maintain uniformity in representation of data.

3.   For the qPCR, it indicates that the lumber spinal cord was removed for mRNA extraction in the Method section, however, the figure 5 legend and result section described the mRNA in the spinal dorsal horn.

4.   In Figure 5, the authors have shown CCK-B receptor mRNA expression in spinal cord only, but not DRG. It would be better if the authors show DRG result in Figure 5.

Comments on the Quality of English Language

The grammar and syntax need to be improved. There are grammar/syntax errors throughout the manuscripts. For example (but not limited to):

 a. In abstract: “Spinal histamine was also caused by the development of allodynia”

 b. In method section: “Each reagent administered into the spinal subarachnoid space dissolved and diluted in artificial cerebrospinal fluid (CSF: Tocris) was administered at a rate of 5 μL/mouse by inserting a 29-gauge needle (Hoshiseido Medical Instrument Industry, Tokyo, Japan) at-tached to a 50 μL micro syringe (Hamilton, Nevada, USA) into the intrathecal space be-tween lumbar vertebrae Nos. 5 and 6 under anesthesia”

c. In result section: “Comparison of pain thresholds at 60 min post-administration by concentration showed no increase in pain threshold at low concentrations (1:100), but at 1:25 and 1:50, pain thresholds returned to the pain threshold value before streptozotocin administration”